

# Seasonal and spatial variations in aerosol vertical distribution and optical properties over China from long-term satellite and ground-based remote sensing

Pengfei Tian[1, 2], Xianjie Cao[1], Lei Zhang[1*], Naixiu Sun[1], Lu Sun[2], Timothy Logan[2], Jinsen Shi[1], Yuan
Wang[3], Yuemeng Ji[2,4],Yun Lin[2], Zhongwei Huang[1], Tian Zhou[1],Yingying Shi[1], Renyi Zhang[2*]

[1]Key Laboratory for Semi-Arid Climate Change of the Ministry of Education, College of Atmospheric Sciences, Lanzhou
University, Lanzhou 730000, China

[2]Department of Atmospheric Sciences, Texas A&M University, College Station, Texas 77843, USA

[3]Jet Propulsion Laboratory, California Institute of Technology, Pasadena, California 91125, USA

[4]Institute of Environmental Health and Pollution Control, School of Environmental Science and Engineering, Guangdong
University of Technology, Guangzhou 510006, China

*Correspondence to*: L. Zhang (zhanglei@lzu.edu.cn) and R. Zhang (renyi-zhang@tamu.edu)

**Abstract.** The vertical distribution and optical properties of aerosols over China are studied using long-term satellite observations from the Cloud–Aerosol Lidar with Orthogonal Polarization (CALIOP) and ground-based lidar observations and the Aerosol Robotic Network (AERONET) data. The CALIOP products are validated using the ground-based lidar measurements at the Semi-Arid Climate and Environment Observatory of Lanzhou University (SACOL). The Taklimakan Desert and Tibetan Plateau regions exhibit the highest depolarization and color ratios because of the natural dust origin, whereas the North China Plain, Sichuan Basin and Yangtze River Delta show the lowest depolarization and color ratios because of aerosols from secondary formation of the anthropogenic origin. Certain regions, such as the North China Plain in spring and the Loess Plateau in winter, show intermediate depolarization and color ratios because of mixed dust and anthropogenic aerosols. In the Pearl River Delta region, the



depolarization and color ratios are similar to but higher, respectively, than those of the other polluted

regions because of combined anthropogenic and marine aerosols. Long-range transport of dust in the

middle and upper troposphere in spring is well captured by the CALIOP observations. The seasonal

variations in the aerosol vertical distributions reveal efficient transport of aerosols from the atmospheric

boundary layer to the free troposphere because of summertime convective mixing. The aerosol extinction

lapse rate in autumn and winter are more positive than those in spring and summer, indicating trapped

aerosols within the boundary layer because of more stable meteorological conditions. More than 80% of

the column aerosols are distributed within 1.5 km above the ground in winter, when the aerosol extinction

lapse rate exhibits a maximum seasonal average in all study regions except for the Tibetan Plateau. The

aerosol extinction lapse rates in the polluted regions are higher than those of the less polluted regions,

indicating a stabilized atmosphere by absorbing aerosols in the polluted regions. Our results reveal that

the satellite and ground-based remote sensing measurements provide the key information on the long-

term seasonal and spatial variations in the aerosol vertical distribution and optical properties, regional

aerosol types, long-range transport, and atmospheric stability, and these data can be utilized to more

precisely assess the direct and indirect aerosol effects on weather and climate.

## 1 Introduction

Atmospheric aerosols affect the radiative budget of the earth–atmosphere system by direct interaction

with solar radiation through scattering and absorption (Boucher et al., 2013; He et al., 2015; Peng et al.,

2016). Also, by acting as cloud condensation nuclei (CCN) or ice nuclei (IN), aerosols alter cloud

formation, albedo, lifetime, precipitation efficiency, and lightning activity, indirectly influencing weather



and climate (Nesbitt et al., 2000; Orville et al., 2001; Li 2008; Yuan et al., 2008; Li et al., 2011; Wang et al., 2011; Rosenfeld et al., 2014; Wang et al., 2014). Currently, the understanding of the aerosol effects remains uncertain, since representation of the aerosol and cloud processes by atmospheric numerical models is difficult, leading to the largest uncertainty in climate projections (Zhang et al., 2007; Boucher

et al., 2013; Wu et al. 2016). Also, absorbing aerosols heat the air and stabilize the atmosphere, exerting a negative impact on air quality (Wang et al., 2013a; Ding et al., 2016; Peng et al., 2016). Furthermore, aerosols provide surfaces for heterogeneous reactions that play a central role in the particles growth, transformation, and properties (Zhang et al., 1994; Zhang et al., 1996; Zhao et al., 2006).

The lack of information on the vertical distributions of aerosols is one of the main underlying factors for

uncertainties in the aerosol direct radiative forcing, since the predictions from atmospheric models typically suffers from large variability (Huneeus et al., 2011). Lidar is a useful tool to provide the vertical distribution of atmospheric aerosols (Sugimoto and Huang, 2014), including ground-based lidars, aircraft-based lidars, and the Cloud–Aerosol Lidar with Orthogonal Polarization (CALIOP) onboard the Cloud–Aerosol Lidar and Infrared Pathfinder Satellite Observation (CALIPSO) satellite (Winker et al., 2009).

The CALIOP observations are consistent with the Moderate Resolution Imaging Spectroradiometer (MODIS) results for the geographical patterns and seasonal variations (Yu et al., 2010). However, the CALIOP aerosol optical depth (AOD) presents an underestimation because of the challenge of the thin-layer detection. The CALIOP AOD over China has been validated using MODIS and the Aerosol Robotic Network (AERONET) data (Liu et al., 2014). The climatological extinction profiles obtained by CALIOP

and the European Aerosol Research Lidar Network (EARLINET) are consistent, despite the issue of a small underestimation (Papagiannopoulos et al., 2016). However, there exist few studies focusing on



validating the CALIOP observed aerosol vertical distributions over China, especially on the climatology of the seasonal vertical profiles of the aerosol extinction coefficient.

The seasonal aerosol vertical distribution over China has been studied using ground-based lidar observations at several sites (He et al., 2008; Huang et al., 2008a; Wu et al., 2011; Cao et al., 2013). The three-dimensional structure of aerosols over China has been evaluated using the frequency of aerosol occurrence derived from CALIOP observations (Guo et al., 2016a). However, the seasonal aerosol extinction coefficient profiles in representative regions over China have not yet been studied. The diverse natural and anthropogenic aerosol sources as well as the geographical and meteorological conditions and transport pathways make China a unique natural laboratory for examination of seasonal dust particles, anthropogenic pollution, and aerosols of mixed types (Zhang et al., 2015). For example, spring dust particles originating from the source regions in the northwest of China are transported to the middle and upper troposphere (Wu et al., 2011; Wang et al., 2013b) and to the downstream regions (Logan et al., 2010; Huang et al., 2015a). Long-range transported particles are typically internally or externally mixed with other aerosol constituents along their transport pathway (Logan et al., 2013; Pan et al., 2015), because of atmospheric processing (Zhang and Zhang, 2005; Zhang et al., 2008). The high contribution of secondary aerosols represents a major characteristic during haze events in China (Guo et al., 2014; Zhang et al., 2015), because of efficient photochemical and particle-phase reactions of organic and inorganic spices (e.g., Lei et al., 2001; Zhang et al., 2002; Suh et al., 2003; Yue et al., 2010). Also, hygroscopic aerosols increase AOD at higher relative humidity (Qiu and Zhang, 2012; Qiu and Zhang, 2013). The vertical distribution of aerosols is governed by transport, which is related to atmospheric stability. For example, effective convection in summer transports aerosols from the planetary boundary layer to the





free troposphere (He et al., 2008; Cao et al., 2013), but stable atmospheric conditions in winter contribute to higher air pollution accumulation near the surface (Zhang et al., 2008). Also, air pollution is further enhanced by the aerosol–planetary boundary layer feedback in China (Peng et al., 2016; Petäjä et al., 2016). The seasonal aerosol mass size distribution over China has been found to be bimodal lognormal

5 by using Nine-stage Anderson Sampler, with a maximum coarse mode in spring and a maximum fine mode in winter (Xin et al., 2015). In situ aerosol composition measurements over 16 urban and rural sites across China have suggested that the seasonal maximum concentrations of most aerosol species occur in winter, whereas the seasonal maximum concentrations of dust aerosol occur in spring (Zhang et al., 2012).

The lidar-observed aerosol depolarization and color ratios are the key parameters in aerosol and cloud

10 characterizations (Sugimoto et al., 2002; Zhou et al., 2013). The color ratio (or wavelength ratio), defined as the ratio between 1064 nm to 532 nm backscatter, is positively related to the aerosol size (Sasano and Browell, 1989). The backscattering linear depolarization ratio is defined as the ratio between the perpendicular and parallel backscatter intensities, and the ratio is zero for spherical aerosols and larger for non-spherical aerosols. The depolarization ratio is used as an aerosol subtyping parameter in the

15 CALIPSO classification algorithm (Omar et al., 2009). The seasonal column-integrated aerosol depolarization ratio in the Loess Plateau region over China has been studied using ground-based depolarization lidar observations at the Semi-Arid Climate and Environment Observatory of Lanzhou University (SACOL) (Tian et al., 2015). CALIOP has continuously conducted observations of the global atmosphere aerosol vertical distribution since June 2006 (Winker et al., 2009).

In this study, we have investigated the regional climatological aerosol vertical distributions and optical properties over eight representative regions in China. Our study focuses on the seasonal aerosol vertical extinction profiles on a regional scale and the seasonal optical properties of dust particles, anthropogenic aerosols, and aerosols of mixed types. We also examine the interaction between aerosols and atmospheric

stability by analyzing the aerosol extinction lapse rate. The study regions, observation sites, instruments, data processing and validating are described in section 2. The spatial distributions of seasonal column AOD are presented in section 3. The seasonal aerosol optical properties and vertical distributions are analyzed and discussed in sections 4 and 5.

## 2 Data and methodology

**2.1 Study regions**

In our study, eight study regions are selected to represent the diverse aerosol types in China (Fig. 1 and Table 1). The Taklimakan Desert region is dominated by dust particles year round (Ge et al., 2014). In the Tibetan Plateau, aerosols are mainly transported from the Taklimakan Desert during spring and summer (Liu et al., 2008; Jia et al., 2015). The Loess Plateau region is dominated by dust particles in

spring, anthropogenic aerosols in summer, and the mixtures of dust with anthropogenic pollution in winter (Wang et al., 2013b). The Northeast China Plain is one of the cleanest regions in China, because it presents less natural dust and anthropogenic pollution (Luo et al., 2014). The Sichuan Basin, North China Plain, and Yangtze River Delta are dominated by anthropogenic pollution (Huang et al., 2011; Zhang et al., 2012). Also, the North China Plain contains anthropogenic dust year round and transported natural dust

in spring (Logan et al., 2013; Huang et al., 2015b). The air quality and pollution dispersal over the Pearl



River Delta are controlled by specific meteorological conditions, and the Pearl River Delta aerosols are dominated by anthropogenic pollution and a small fraction of marine aerosols (Xu et al., 2015).

## 2.2 CALIOP data and processing

A comparison of the CALIOP observations with the MODIS products suggests that the CALIPSO version

3 products provide a consistent and representative mean regional and seasonal aerosol load and distribution compared with the version 2 products (Koffi et al., 2012). The CALIOP version 3 level 2 aerosol and cloud products from June 2006 to January 2016 are employed in this study. All of the results in this study are under cloud-free conditions, i.e., no cloud layer in the 5 km cloud layer products. The parameters of the aerosol layers, such as layer-integrated aerosol color ratio, layer-integrated aerosol

depolarization ratio, and layer top and base altitudes, are derived from the CALIOP 5 km aerosol layer products. The column aerosol AOD is also derived from the CALIOP 5 km aerosol layer products, and the average seasonal AOD is calculated using the following quality control procedures: (1) cloud free; (2) $0 \leq AOD_{532nm} \leq 3.0$ ; (3) $-100 \leq CAD\_Score \leq -20$ ; (4) $Ext\_QC = 0, 1$ ; and (5) $0 < AOD_{532nm,unc}/AOD_{532nm} \leq 100\%$, where $AOD_{532nm}$ is the aerosol optical depth at 532 nm wavelength,

$CAD\_Score$ is the cloud-aerosol discrimination score, $Ext\_QC$ is the extinction quality control flags, and $AOD_{532nm,unc}$ is the uncertainty of $AOD_{532nm}$. The seasonal aerosol extinction vertical profiles are derived from the CALIOP 5 km aerosol profile products with similar quality control procedures as in Winker et al. (2013): (1) $-100 \leq CAD\_Score \leq -20$; (2) $Ext\_QC = 0, 1$; (3) fill values representing clear sky conditions are assigned an extinction value of 0.0 km$^{-1}$; (4) range bins with uncertainty of 99.9

km$^{-1}$ and bins at lower altitudes in the profile are rejected; and (5) extinction values near the surface less than $-0.2$ km$^{-1}$ are ignored. Higher thresholds are adopted for the CALIOP data processing during

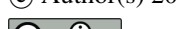


daytime hours than nighttime hours because of the daytime solar background illumination (Winker et al.,

2013). Thus, weakly scattering aerosol layers that are detected at night may not be detected during the

daytime. Therefore, the averaged daytime extinction profiles are higher and tend to be noisier than the

nighttime profiles. Consequently, the nighttime CALIOP aerosol profiles and layer products are used in

this study.

Similar to the temperature lapse rate, the aerosol extinction (coefficient) lapse rate ( $\gamma_{ext}$ ) is defined as,

$$\gamma_{ext} = -d\sigma(z)/dz, \qquad (1)$$

where $\sigma(z)$ is the extinction coefficient at the height of $z$. The unit of $\gamma_{ext}$ is km$^{-2}$ (km$^{-1}$/km). Stable

meteorological conditions are unfavorable for aerosol vertical transport (Kipling et al., 2016) and lead to

a high positive aerosol extinction lapse rate. The extinction lapse rate is more applicable to a

climatological aerosol vertical profile rather than an instantaneous profile, because an elevated aerosol

layer in the instantaneous profile leads to a negative aerosol extinction lapse rate. The ratio of AOD with

1.5 km above the ground to the column AOD, $R_{AOD,1.5km}$, is derived from the average extinction profiles:

$$R_{AOD,1.5km} = \sum_{k=base}^{base+1.5} \sigma(z_k) \Big/ \sum_{k=base}^{top} \sigma(z_k), \qquad (2)$$

**2.3 Ground-based lidar and extinction retrieval**

The Semi-Arid Climate and Environment Observatory of Lanzhou University (SACOL, 35.946 °N,

104.137 °E, and 1965.8 m ASL) is an international research observatory located in the semi-arid region

of the Loess Plateau in northwest China (Huang et al., 2008b). Lidar observations were performed by a

NIES (National Institute for Environmental Studies) depolarization lidar (Huang et al., 2010) from





October 2009 to August 2012. The lidar data is denoised using the empirical mode decomposition (EMD)-based method according to Tian et al. (2014). The lidar equation (Fernald, 1984) is as follows:

$$P(z) = ECz^{-2}[\beta_1(z) + \beta_2(z)]T^2(z), \tag{3}$$

where $P(z)$ is the lidar backscattering return signal at the height of $z$, $E$ is an output energy monitor

pulse, $C$ is a calibration constant, $\beta_1(z)$ is the aerosol backscattering coefficient, $\beta_2(z)$ is the molecule

backscattering coefficient, and $T(z) = \exp\{-\int_0^z[\sigma_1(z) + \sigma_2(z)]dz\}$ is the transmittance. $\sigma_1(z)$ is the

aerosol extinction coefficient, and $\sigma_2(z)$ is the molecule extinction coefficient. The ratio between $\sigma_1(z)$

and $\beta_1(z)$ (known as the lidar ratio or extinction to backscattering ratio) is pre-assigned to solve the

equation, because this equation is not closed due to the two unknowns $\sigma_1(z)$ and $\beta_1(z)$. The retrieved

aerosol extinction coefficients suffer from large uncertainties because of the pre-assigned lidar ratios. If

the AOD is simultaneously observed using a sun photometer, the aerosol extinction coefficient profile

can be retrieved using the AOD-constrained Fernald (1984) method, as introduced by Huang et al. (2010).

The aerosol extinction coefficients retrieved from the AOD-constrained retrieval method is subjected to

less uncertainty because the lidar ratio assumption is not required.

**2.4 AERONET sites and data processing**

The aerosol volume size distribution and single scattering albedo (SSA) data from the Aerosol Robotic

Network (AERONET) are utilized to characterize the typical aerosol types at the SACOL (35.946° N,

104.137° E), Beijing (39.98°N, 116.38°E), and Taihu (31.42°N, 120.22°E) sites (Fig. 1). Data are

available from 28 July 2006 to 10 August 2012 for SACOL, from 9 March 2001 to 23 March 2015 for

Beijing, and from 6 September 2005 to 4 October 2012 for Taihu. The aerosol size distribution and SSA

are the key parameters in aerosol classification (Li et al., 2007). The aerosol classification method by





Logan et al. (2013) is also considered in this study. This method is based on two parameters from the AERONET observations: the Ångström exponent ($\alpha_{440-870}$) and single scattering co-albedo ($\omega_{oabs440}$). The Ångström exponent is a good indicator of the size of aerosols, and a threshold of $\alpha_{440-870} = 0.75$ is used to define fine ($\alpha_{440-870} > 0.75$) and coarse mode ($\alpha_{440-870} < 0.75$) aerosols (Eck et al., 2005).

The single scattering co-albedo is the ratio of absorption to extinction aerosol optical depths. A threshold of $\omega_{oabs440} = 0.07$ is set to define weakly ($\omega_{oabs440} < 0.07$) and strongly ($\omega_{oabs440} > 0.07$) absorbing aerosols. The weakly and strongly absorbing pollution, mineral dust, and biomass burning aerosols are classified according to the method by Logan et al. (2013).

**2.5 Validation of the CALIOP extinction profiles**

The aerosol extinction coefficients in the free troposphere are typically underestimated under clean conditions (Winker et al., 2013). The climatological extinction profiles obtained by CALIOP and the European Aerosol Research Lidar Network (EARLINET) are consistent, although the CALIOP results show a small underestimation (Papagiannopoulos et al., 2016). Validation of seasonal CALIOP aerosol extinction coefficient profiles using ground-based lidar observations at SACOL is carried out in this study.

The nighttime CALIOP observations with a distance of less than 100 km from SACOL are averaged to calculate the seasonal extinction coefficient profiles using the data quality control procedures described in section 2.2. Hourly average NIES lidar extinction profiles are retrieved using the AOD-constrained Fernald method developed by Huang et al. (2010). The seasonal extinction profiles are derived from the hourly averages. The seasonal vertical distributions are well captured by the CALIOP observations (Fig.

2). The NIES lidar spring extinction profile is very close to that observed by a Micro-Pulse Lidar (MPL) at SACOL in the spring of 2007 (Huang et al., 2008a). The seasonal aerosol extinction profiles over

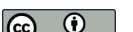



SACOL have been studied using the observations from a CE370-2 lidar (Cao et al., 2013). However, because observations under dusty conditions were excluded from the averages in Cao et al. (2013), the seasonal variations of the CE370-2 lidar extinction profiles in the boundary layer are inconsistent with the seasonal variations of the NIES lidar extinction profiles presented here.

**3 Spatial distribution of aerosol optical depth**

The combined daytime and nighttime seasonal average AOD over China from the CALIOP observations is shown in Fig. 3. The CALIOP AOD is consistent with the MODIS AOD (Luo et al., 2014; Tao et al., 2015), MISR (Multi-angle Imaging Spectroradiometer) AOD (Qi et al., 2013), and ground-based AOD (Che et al., 2015) with regard to the geographical patterns and seasonal variations. CALIOP provides a

full coverage of China, including the Tibetan Plateau and Taklimakan Desert regions with a $1.0° \times 2.5°$ latitude-longitude grid, which is an important advantage over the passive satellites. The seasonal AOD hotspots over the Taklimakan Desert, North China Plain, Sichuan Basin, and Yangtze River Delta are clearly evident in the CALIOP observations. The AOD hotspots over China coincide with high $PM_{2.5}$ (particles with the aerodynamic diameter smaller than 2.5 μm) concentrations (Zhang and Cao, 2015),

which are also associated with population hotspots over China (Ma et al., 2016), except in the Taklimakan Desert region.

Dust represents the main natural aerosol type over northwest China, especially in spring (Wang et al., 2013b). In situ measurements over 16 sites across China have revealed that 35% of the aerosols are composed of mineral dust (Zhang et al., 2012). High ratios of the dust-only AOD to the total AOD are

centered on the dust source regions in northwest China (Fig. S1 a). The ratio of the dust-only to total AOD





is also high over the Tibetan Plateau, because of transported dust from the Taklimakan Desert (Liu et al., 2008; Jia et al., 2015). Mixed dust with anthropogenic pollution/biomass burning aerosols are classified as polluted dust in the CALIPSO aerosol subtyping algorithm (Omar et al., 2009). A hot spot of the polluted-dust AOD to total AOD ratio is in the North China Plain (Fig. S1 b). The polluted dust in the

5  North China Plain is considered to be anthropogenic dust (Huang et al., 2015b).

## 4 Aerosol optical properties over the representative regions

The lidar-observed aerosol depolarization ratio and color ratio are the key optical parameters in aerosol characterization. The CALIOP layer-integrated volume depolarization ratio ($\delta'_{layer}$) and layer-integrated volume color ratio ($\chi'_{layer}$) are calculated from:

$$\delta'_{layer} = \sum_{k=top}^{base} \beta_{532,\perp}(z_k) / \sum_{k=top}^{base} \beta_{532,\parallel}(z_k), \qquad (4)$$

$$\chi'_{layer} = \sum_{k=top}^{base} \beta_{1064,k} / \sum_{k=top}^{base} \beta_{532,k}, \qquad (5)$$

where $\beta_{532,\perp}(z_k)$ and $\beta_{532,\parallel}(z_k)$ refer to the polarized and depolarized attenuated backscatter signals, respectively, and $\beta_{1064,k}$ and $\beta_{532,k}$ refer to the attenuated backscatter coefficients at 1064 and 532 nm wavelengths, respectively. The layer-integrated volume depolarization ratio $\delta'_{layer}$ and layer-integrated

15  volume color ratio $\chi'_{layer}$ are then corrected to the layer-integrated aerosol (or particle) depolarization ratio $\delta_{layer}$ and layer-integrated aerosol (or particle) color ratio $\chi_{layer}$ (Omar et al., 2009). Dust particles are composed of non-spherical, coarse-mode-dominated mineral dust (Kandler et al., 2011), while anthropogenic pollution aerosols are fine-mode-dominated particles with a spherical shape (Sugimoto et al., 2002; Omar et al., 2005). The dust particles have a volume depolarization ratio of higher than 0.2,

20  while anthropogenic pollution aerosols have a volume depolarization ratio of lower than 0.1 (Xie et al.,





2008; Nemuc et al., 2013). Dust particles are internally or externally mixed with other aerosol types along their transport pathway (Pan et al., 2015). Marine aerosols are dominated by sea salts, which are coarse-mode-dominant but smaller than the desert dust (Porter and Clarke, 1997). The color ratios of sea salt aerosols are higher than those of sulfate aerosols (Sugimoto et al., 2002), and the aerosol depolarization

ratios for marine aerosols range from 0.01 to 0.03 (Groß et al., 2011).

The scatter plots for layer-integrated aerosol color ratios versus layer-integrated aerosol depolarization ratios for the eight study regions are shown in Fig. 4. In order to better compare aerosol optical properties in different study regions, the ratio of the data point number in a 0.067×0.020 color ratio-depolarization ratio grid to the maximum data point number in a grid in each region, which is referred to as grid data

point number ratio hereafter, is depicted by color in Fig. 4. The green, yellow and red data points, which have a grid data point number ratio of 0.4 to 1.0, account for more than 85% of the total data points. Dust-dominated aerosols are scattered in the upper right area in Figs. 4 (a) and (b) (i.e., in the Taklimakan Desert and Tibetan Plateau regions), corresponding to large and non-spherical particles. In contrast, anthropogenic aerosols dominated by secondary formation are scattered in the lower left area in Figs. 4

(d), (e), (f) and (g) (i.e., in the Northeast China Plain, Sichuan Basin, North China Plain, and Yangtze River Delta regions), corresponding to small and spherical particles. For the Loess Plateau region, the data points are scattered from the lower left all the way to the upper right, because of mixed particles from anthropogenic pollution and natural dust. The data points in the Pearl River Delta show the similar scattered pattern, but with a higher color ratio than those of anthropogenic pollution aerosols, because of

the existence of a fraction (about 20%) of larger-sized marine sea salt aerosols (Xu et al., 2015).

To better understand the optical proprieties for the different aerosol types and their mixtures, the seasonal average layer-integrated aerosol color ratios versus the layer-integrated aerosol depolarization ratios of the eight representative study regions are present in Fig. 5. The seasonal scatter plots for the layer-integrated aerosol color ratios versus layer-integrated aerosol depolarization ratios for the eight study

regions are shown in Figs. S2-S9. The spring regional average depolarization ratios are higher than those of the other seasons in the same region and higher than 0.1, except in the Pearl River Delta region. The summer regional average depolarization ratios are lower than those during the other seasons in the same region and lower than 0.1, except for the regions in northwest China, i.e., the Taklimakan Desert, Tibetan Plateau, and Loess Plateau. All of the seasonal average data points of the Taklimakan Desert and Tibetan

Plateau are scattered in the upper right area (with large and non-spherical particles), whereas those of the Northeast China Plain and Pearl River Delta are scattered in the lower left area (with small and spherical particles). For the regions dust plays an important role in spring and anthropogenic pollution dominates in summer, i.e., the Loess Plateau, North China Plain, Sichuan Basin, and Yangtze River Delta, the data points are scattered along the regression line from the lower left to the upper right in the sequence of

summer, autumn, winter and spring. The depolarization ratio differences between the spring and summer averages for these four regions are in the range of 0.11-0.12.

The long-term optical properties and characterization of aerosols are also available at three AERONET sites, SACOL, Beijing and Taihu, over China. The AERONET observations provide a validation of the CALIOP characterization of the seasonal aerosols. The dust-dominant aerosols of spring SACOL,

anthropogenic pollution-dominated aerosols of summer Beijing and summer Taihu, and aerosols of mixed type of spring Beijing are clearly represented by AERONET observations (Figs. 6 and 7). Most of the



spring aerosols over SACOL are of large size ($\alpha_{440-870} < 0.75$) and strongly absorbing ($\omega_{oabs440} >$ 0.07) (Fig. 7). The natural dust-dominated SACOL aerosols in spring are mainly in the coarse mode and present an increasing spectral SSA trend (Fig. 6). Aerosols are dominated by anthropogenic pollution during the summer in Beijing and Taihu, with a relatively higher fine mode peak in the size distribution

and a decreasing spectral SSA trend. Aerosols at the Beijing site are more absorbing, with a relatively higher coarse mode and lower fine mode than those at the Taihu site. Aerosols in Beijing during spring are of the mixed type, which are dominated by dust and anthropogenic aerosols with high absorption. Therefore, the spring Beijing aerosols have a similar coarse mode and a higher fine mode than those of the spring SACOL aerosols. In addition, the spring Beijing aerosols exhibit a spectral SSA trend that

differs from both dust and pollution aerosols.

## 5 Aerosol vertical distributions over the representative regions

As a major characteristic of aerosols over China, spring dust is transported to the middle and higher troposphere, which is well reflected from the CALIOP observations on a regional scale (Fig. 8). Strong vertical mixing in summer transports more aerosols from the atmospheric boundary layer to the free

troposphere, including the Taklimakan Desert dust. In contrast, stable meteorological conditions in autumn and winter trap more aerosols within the boundary layer. About 80% of the column aerosols in winter are distributed within 1.5 km above the ground (Table 2), and the extinction lapse rates (Eq. 1) increase to over 0.15 km$^{-2}$ (Fig. 9).

To better understand the aerosol properties in the Taklimakan Desert region, the CALIOP detected

number and depth of the aerosol layers with a layer base within 2 km above the ground from June 2006

to January 2016 are calculated. There are 6904, 12727, 19445 and 14510 aerosol layers, with a layer depth (average $\pm$ standard deviation) of 2.464 $\pm$ 1.107 km, 2.396 $\pm$ 1.336 km, 1.705 $\pm$ 1.014 km, and 0.960 $\pm$ 0.536 km in spring, summer, autumn and winter, respectively. In the Taklimakan Desert region, spring dust aerosols show the highest seasonal average depolarization ratio of 0.32 $\pm$ 0.08 (Fig. 5), the highest

layer depth, and the largest extinction coefficients (Fig. 8 a). Dust is efficiently transported to the upper troposphere in summer (also in Ge et al., 2014), and more aerosol layers are detected in autumn than the other seasons. Winter dust aerosols are trapped within the boundary layer and mixed with anthropogenic pollution, with a thinner layer depth and a lower average depolarization ratio of 0.23 $\pm$ 0.10. In addition, 89% of the total column aerosols are distributed within 1.5 km above the ground in winter (Table 2).

The Tibetan Plateau is a clean region with low anthropogenic aerosol loading, but Taklimakan Desert dust can be transported to the Tibetan Plateau in spring and summer (Liu et al., 2008; Jia et al., 2015). The spring and summer extinction profiles of the Tibetan Plateau aerosols are much larger than the autumn and winter profiles. CALIOP-detected nighttime aerosol layer numbers over the Tibetan Plateau are 16502, 11579, 6667 and 8030 in spring, summer, autumn and winter, respectively. The maximum

spring and summer average extinction coefficients are approximately 0.017 km$^{-1}$ at 5.0 km height, whereas the maximum autumn and winter coefficients are less than 0.005 km$^{-1}$. Note that the extinction coefficients may be overestimated in the Tibetan Plateau, because the weakly scattering aerosol layers may not be detected by CALIOP.

Aerosols are mainly trapped within the boundary layer in autumn and winter over the Loess Plateau region

(Table 2 and Fig. 8 c). Transported spring dust causes higher extinction coefficients in the middle and

upper troposphere. Summer extinctions are larger than those for the other seasons in the Loess Plateau region, which may be attributable to more hygroscopic aerosols due to more abundant water vapor and higher temperatures in summer (Su et al., 2014). The seasonal aerosol vertical distributions over the Northeast China Plain region (Fig. 8 d) are similar to that of the Loess Plateau but with lower extinctions

because of both lower natural dust and lower anthropogenic aerosol loadings (Luo et al., 2014).

The Sichuan Basin, North China Plain, and Yangtze River Delta regions contain high levels of anthropogenic pollution, and the aerosol extinctions are higher than those of the spring Taklimakan Desert dust (Fig. 8). High anthropogenic emissions, efficient secondary aerosol formation (Zhang et al., 2015), and stable meteorological conditions (Miao et al., 2015) contribute to large aerosol loadings within the

atmospheric boundary layer in these regions. The aerosol extinctions within the atmospheric boundary layer are large in summer and winter for the North China Plain and Yangtze River Delta, whereas the values for the Sichuan Basin are relatively low in summer. The $SO_2$ and $NO_2$ concentrations over the Sichuan Basin are lower than those over the North China Plain and Yangtze River Delta regions (Wang et al., 2015; Cui et al., 2016), and the Sichuan Basin region is also correspond to fewer sunny days (Liu

et al., 2010), leading to low photochemical activity. Moreover, strong vertical mixing in summer also transports aerosols vertically in the Sichuan Basin region.

Although local anthropogenic pollution plays a major role in the Pearl River Delta region, the northwest winter monsoon transports continental aerosols, and the southeast summer monsoon transports marine aerosols to this region (Wu et al., 2013). The aerosol extinction coefficients within the planetary boundary

layer in autumn and winter are much higher than those in spring and summer (Fig. 8 h). A lower planetary





boundary layer height in autumn and winter (Guo et al., 2016b) also contributes to higher aerosol loading near the surface. The anti-cyclone high-pressure systems and sea breeze induce three inversion layers in the Pearl River Delta region (Fan et al., 2008), which are responsible for two aerosol layers in the vertical direction (Ansmann et al., 2005).

Convective transport has been suggested to be an important factor that controls the vertical distribution of aerosols (Kipling et al., 2016). It has been suggested that absorbing aerosols (including black carbon) play an important role in determining the atmospheric stability (Wang et al., 2013a; Peng et al., 2016). Light absorption and scattering of the atmospheric aerosols heat the air and decrease the surface temperature, enhancing accumulation of air pollution (Ding et al., 2016; Petäjä et al., 2016). The

absorption aerosol optical depth (AAOD) over the polluted regions (i.e., the Sichuan Basin, North China Plain, Yangtze River Delta, and Pearl River Delta) is much higher than the other regions in China (Gustafsson and Ramanathan, 2016). The extinction lapse rates over the polluted regions are higher than the less polluted regions (Fig. 9). The extinction lapse rates are higher than $0.2~km^{-2}$ in the polluted regions, while those in the less polluted regions are generally lower than $0.1~km^{-2}$. The autumn and winter

extinction lapse rates are higher than those of the spring and summer rates for most regions, explainable by a lower atmospheric boundary layer height (Guo et al., 2016b) and a higher fraction of black carbon aerosols (Schleicher et al., 2013) in autumn and winter than those in spring and summer. The extinction lapse rate in the Taklimakan Desert region shows a seasonal maximum in winter, when the planetary boundary layer height is low (Guo et al., 2016b) and elevated black carbon aerosols from coal combustion

for heating in winter. The spring extinction lapse in the Taklimakan Desert region is higher than those in summer and autumn, probably attributable to absorbing dust aerosols in spring.



**6 Conclusions**

The vertical aerosol distributions and optical properties are essential in assessing the aerosol direct and indirect radiative forcing, but few studies have reported these regional climatological data over China using combined long-term satellite and ground-based remote sensing observations. In this work, the

CALIOP satellite products are validated using the ground-based lidar observations, and the CALIOP seasonal AOD spatial distribution is obtained. The CALIOP aerosol layer products and AERONET data are employed to evaluate the aerosol optical properties of the dust-dominated particles, anthropogenic pollution-dominated aerosols, and aerosols of the mixed types. The CALIOP aerosol profile products are used to study the seasonal and spatial variations in the aerosol extinction coefficients for eight

representative regions over China.

The seasonal variations in the aerosol vertical distributions are well captured by the CALIOP observations, although the CALIOP aerosol extinctions represent an underestimation when compared with the ground-based lidar results at SACOL. The long-term column AOD and aerosol vertical distribution over the Tibetan Plateau, which are typically difficult to obtain by passive satellites, are determined using the

CALIOP observations. The AOD hotspots over China are consistently co-located with the hotspots of high $PM_{2.5}$ concentrations and population, except in the Taklimakan Desert region.

The dust-dominant Taklimakan Desert and Tibetan Plateau regions exhibit the highest depolarization ratios and the highest color ratios, whereas the anthropogenic pollution-dominated North China Plain, Sichuan Basin and Yangtze River Delta regions show the lowest depolarization ratios and the lowest color

ratios. The spring North China Plain and the winter Loess Plateau show intermediate depolarization and

color ratios because of the mixed natural dust and anthropogenic pollution particles. In the Pearl River Delta region, the depolarization and color ratios are similar to but higher, respectively, than those of the polluted regions because of the combined anthropogenic pollution and marine aerosols.

Long-range transport of dust in the middle and higher troposphere during the spring season is clearly

evident in the CALIOP observed aerosol extinction coefficient profiles. The seasonal variations in aerosol vertical distributions indicate efficient transport of aerosols from the atmospheric boundary layer to the free troposphere because of summertime convective mixing, but stable meteorological conditions trap more aerosols within the boundary layer in autumn and winter. The aerosol extinction lapse rate is closely correlated to the atmospheric stability, with higher values in autumn and winter than spring and summer.

More than 80% of the column aerosols are distributed within 1.5 km above the ground in winter, when aerosol extinction lapse rate reaches a maximum seasonal average in all the study regions except for the Tibetan Plateau. For the polluted regions (i.e., the Sichuan Basin, North China Plain, Yangtze River Delta, and Pearl River Delta), the aerosol extinction lapse rates in the planetary boundary layer are higher than those of the less polluted regions (the Taklimakan Desert, Tibetan Plateau, Loess Plateau, and Northeast

China Plain). Our results suggest that absorbing aerosols may contribute to the high aerosol extinction lapse rates in the heavily polluted regions.

Hence, we have for the first time presented the seasonal and spatial variations of the profiles of aerosol extinction coefficients and identified the dominant regional aerosol types over China, using combined long-term satellite and ground-based remote sensing observations. The vertical aerosol distributions and





optical properties from our work can be utilized to more precisely assess the direct and indirect aerosol effects on weather and climate.

## 7 Data availability

The CALIOP data is available from the National Aeronautics and Space Administration (NASA) site 5 (http://www-calipso.larc.nasa.gov/tools/data_avail/). The NIES lidar data is available from the SACOL site (http://climate.lzu.edu.cn/data/data.asp) upon request. The sun photometer data is available from the AERONET website (http://aeronet.gsfc.nasa.gov/). The regional climatology products in the eight representative regions over China, the lidar profiles at SACOL, and the AERONET results data in this paper are available from the authors upon request. The gridded climatology aerosol extinction coefficient 10 profiles (not shown in this paper) and AOD over China with a $1.0° \times 2.5°$ latitude-longitude grid, which can be used as model input or to test model results, are also available from the authors upon request.



**Acknowledgements.** This research was funded by the National Natural Science Foundation of China (41475008, 41521004, 41225018 and 41405113). P. Tian was supported by the China Scholarship Council as a visiting scholar at Texas A&M University from September 2015 to August 2016. The authors are grateful to the National Aeronautics and Space Administration (NASA) for providing the CALIPSO satellite data used in this study and SACOL for providing the ground-based lidar data. We also thank the AERONET program for its efforts to establish and maintain the SACOL, Beijing, and Taihu sites.

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





**Table 1.** Classification of the study regions

| Region | Abbreviation | Latitude-longitude range |
| --- | --- | --- |
| Loess Plateau | LP | 34.0-38.0° N, 103.0-112.0° E |
| North China Plain | NCP | 34.0-41.0° N, 113.0-119.0° E |
| Northeast China Plain | NEP | 43.0-49.0° N, 120.0-130.0° E |
| Pearl River Delta | PRD | 21.5-25.0° N, 111.0-116.0° E |
| Sichuan Basin | SB | 28.0-33.0° N, 103.0-110.0° E |
| Taklimakan Desert | TD | 37.0-42.0° N, 78.0-88.0° E |
| Tibetan Plateau | TP | 30.0-36.0° N, 80.0-100.0° E |
| Yangtze River Delta | YRD | 28.0-33.0° N, 116.0-122.0° E |




**Table 2.** Ratio of the AOD within 1.5 km height above the ground to the total column AOD (%).

|     | TD   | TP   | LP   | NEP  | SB   | NCP  | YRD  | PRD  | Average |
|-----|------|------|------|------|------|------|------|------|---------|
| MAM | 71.2 | 61.0 | 61.9 | 58.8 | 61.1 | 63.7 | 65.8 | 48.6 | 61.5    |
| JJA | 58.4 | 64.0 | 68.9 | 71.2 | 56.7 | 70.0 | 74.8 | 75.6 | 67.5    |
| SON | 74.5 | 65.6 | 79.7 | 74.3 | 72.9 | 81.3 | 84.4 | 83.0 | 77.0    |
| DJF | 89.0 | 65.9 | 82.0 | 78.1 | 77.5 | 82.8 | 83.2 | 82.8 | 80.2    |



**Figure Captions**

**Figure 1.** Study regions (square boxes) and AERONET sites (triangles) over a 1.0° × 2.5° latitude–longitude gridded surface elevation.

**Figure 2.** Aerosol extinction coefficient profiles from the ground-based NIES lidar and CALIOP observations over SACOL from October 2009 to August 2012: (a) spring; (b) summer; (c) autumn; and (d) winter. Altitudes of CALIOP observations are transferred to heights above the ground level of SACOL. The left and right boundaries of the light grey shadowed area depict the NIES lidar extinction coefficient averages with one standard deviation.

**Figure 3.** Seasonal AOD with a 1.0° × 2.5° latitude-longitude grid over China derived from CALIOP observations from June 2006 to January 2016.

**Figure 4.** Scatter plots of the layer-integrated aerosol color ratios versus the layer-integrated aerosol depolarization ratios for the regions: (a) Taklimakan Desert (TD); (b) Tibetan Plateau (TP); (c) Loess Plateau (LP); (d) Northeast China Plain (NEP); (e) Sichuan Basin (SB); (f) North China

Plain (NCP); (g) Yangtze River Delta (YRD); and (h) Pearl River Delta (PRD).

**Figure 5.** Seasonal average layer-integrated aerosol color ratios versus layer-integrated aerosol depolarization ratios over the eight representative study regions in China.

**Figure 6.** (a) Volume size distribution, (b) spectral single scattering albedo (SSA) for dust-dominant aerosols (SACOL in spring), anthropogenic aerosols (Beijing and Taihu in summer), and

aerosols of mixed type (spring Beijing) derived from the long-term AERONET observations.

**Figure 7.** Classification of the AERONET sites representing dust (SACOL in spring), anthropogenic aerosols (Beijing and Taihu in summer), and aerosols of the mixed types (Beijing in spring).





**Figure 8.** Aerosol extinction coefficient profiles for the following regions: (a) the Taklimakan Desert

(TD); (b) the Tibetan Plateau (TP); (c) the Loess Plateau (LP); (d) the Northeast China Plain

(NEP); (e) the Sichuan Basin (SB); (f) the North China Plain (NCP); (g) the Yangtze River

Delta (YRD); and (h) the Pearl River Delta (PRD).

5  **Figure 9.** Extinction lapse rates within 1.5 km above the ground. For the profiles where the extinction

maximum is not reached at the bottom (such as the profiles for the Sichuan Basin because of the

topography), the extinction lapse rates are calculated within 1.5 km above the height of the

maximum extinctions.





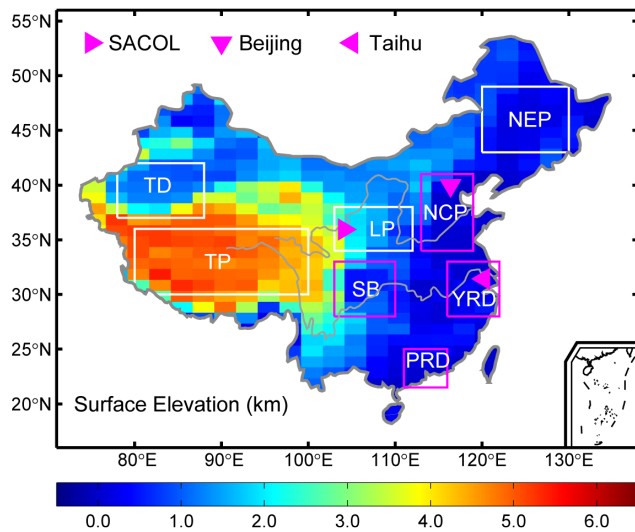

**Figure 1.** Study regions (square boxes) and AERONET sites (triangles) over a 1.0° × 2.5° latitude–longitude gridded surface elevation.





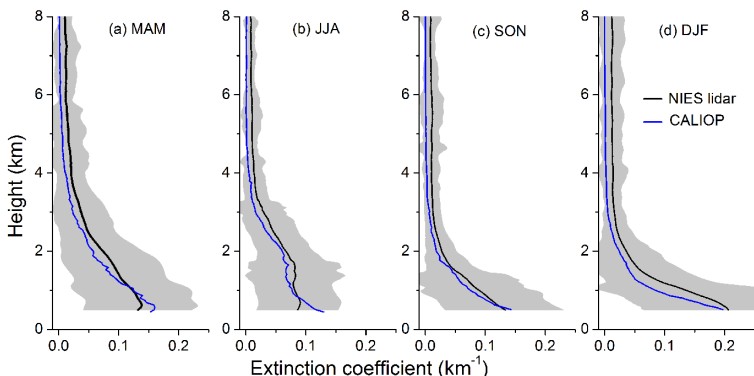

**Figure 2.** Aerosol extinction coefficient profiles from the ground-based NIES lidar and CALIOP observations over SACOL from October 2009 to August 2012: (a) spring; (b) summer; (c) autumn; and (d) winter. Altitudes of CALIOP observations are transferred to heights above the ground level of SACOL. The left and right boundaries of the light grey shadowed area depict
5   the NIES lidar extinction coefficient averages with one standard deviation.





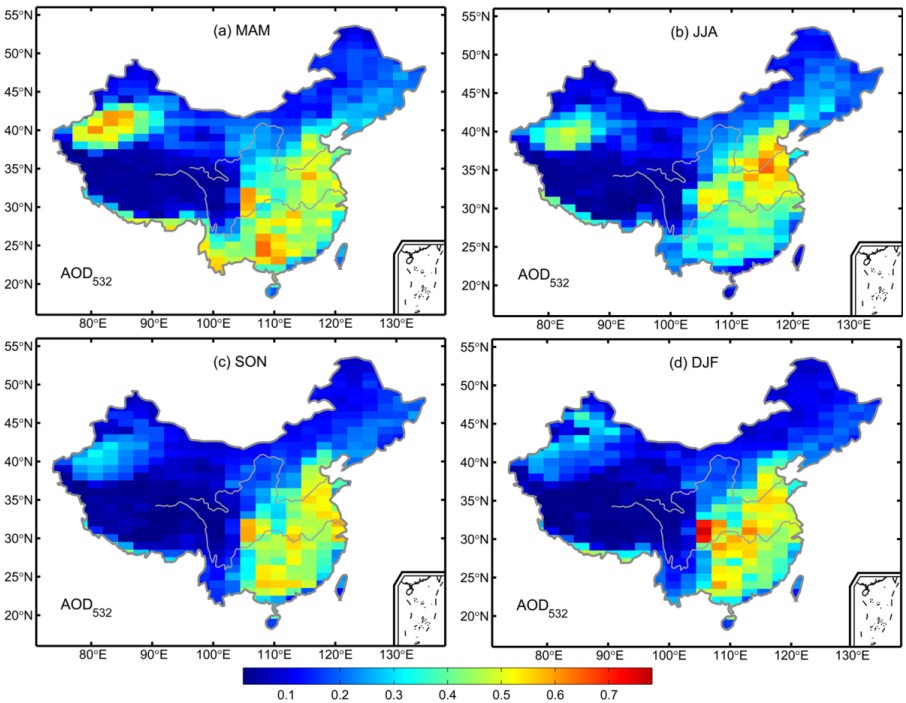

**Figure 3.** Seasonal AOD with a 1.0° × 2.5° latitude-longitude grid over China derived from the CALIOP observations from June 2006 to January 2016.





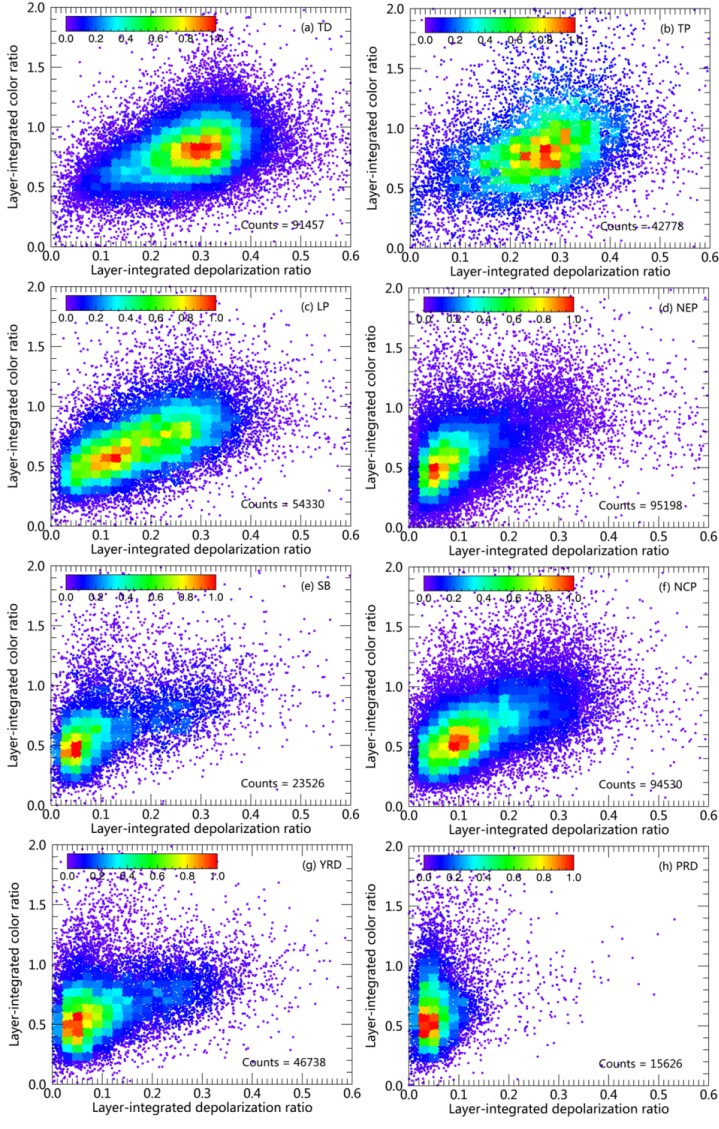

**Figure 4.** Scatter plots of the layer-integrated aerosol color ratios versus the layer-integrated aerosol depolarization ratios for the regions: (a) Taklimakan Desert (TD); (b) Tibetan Plateau (TP); (c) Loess Plateau (LP); (d) Northeast China Plain (NEP); (e) Sichuan Basin (SB); (f) North China Plain (NCP); (g) Yangtze River Delta (YRD); and (h) Pearl River Delta (PRD).





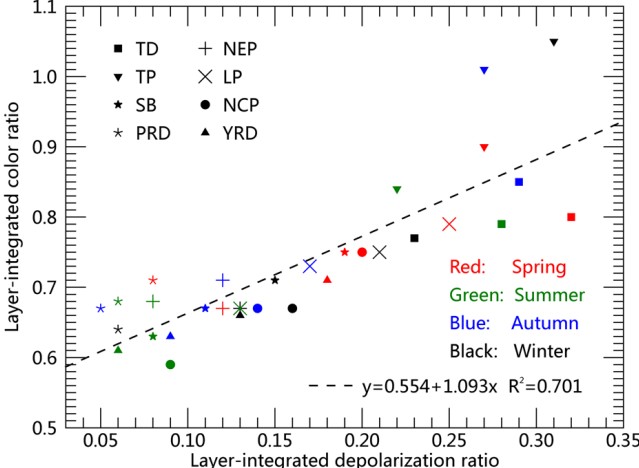

**Figure 5.** Seasonal average layer-integrated aerosol color ratios versus layer-integrated aerosol depolarization ratios over the eight representative study regions in China.



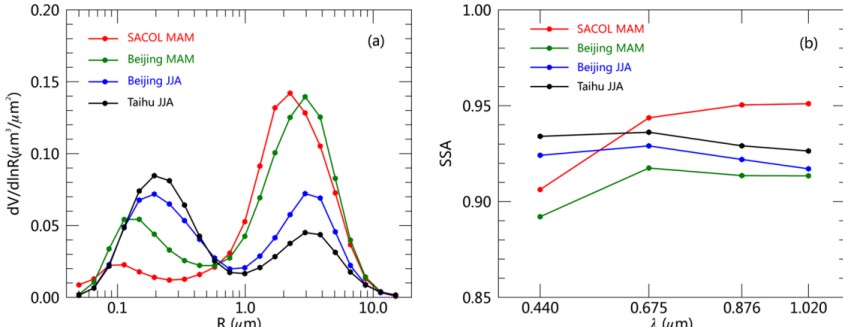

**Figure 6.** (a) Volume size distribution, (b) spectral single scattering albedo (SSA) for dust-dominant aerosols (SACOL in spring), anthropogenic aerosols (Beijing and Taihu in summer), and aerosols of mixed type (spring Beijing) derived from the long-term AERONET observations.



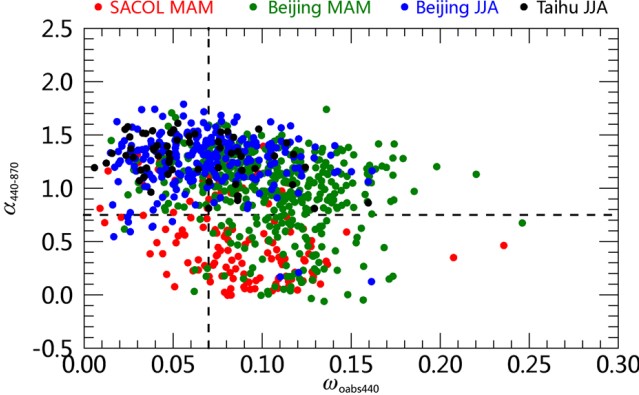

**Figure 7.** Classification of the AERONET sites representing dust (SACOL in spring), anthropogenic aerosols (Beijing and Taihu in summer), and aerosols of the mixed types (Beijing in spring).





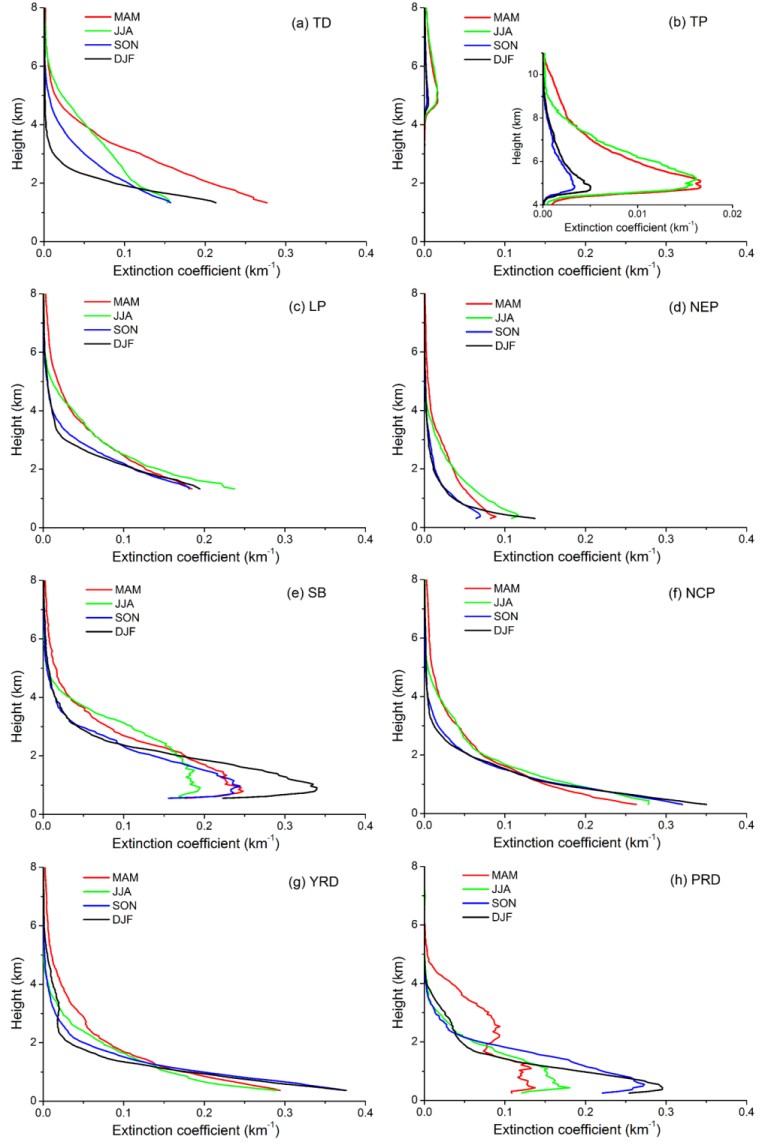

**Figure 8.** Aerosol extinction coefficient profiles for the following regions: (a) the Taklimakan Desert (TD); (b) the Tibetan Plateau (TP); (c) the Loess Plateau (LP); (d) the Northeast China Plain (NEP); (e) the Sichuan Basin (SB); (f) the North China Plain (NCP); (g) the Yangtze River Delta (YRD); and (h) the Pearl River Delta (PRD).



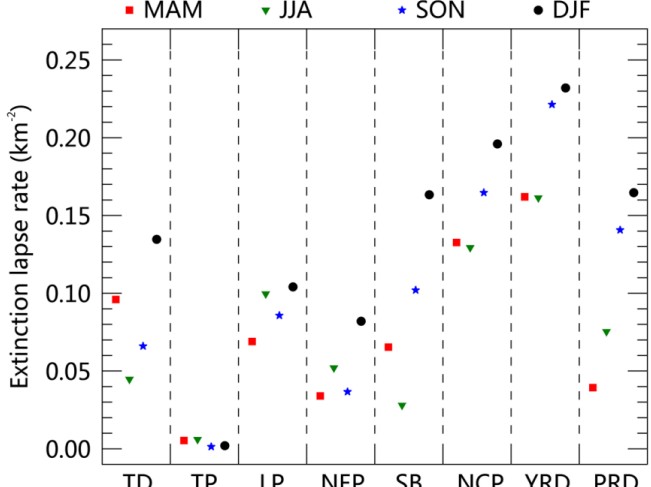

**Figure 9.** Extinction lapse rates within 1.5 km above the ground. For the profiles where the extinction maximum is not reached at the bottom (such as the profiles for the Sichuan Basin because of the topography), the extinction lapse rates are calculated within 1.5 km above the height of the maximum extinctions.

