# Peer review of "Aerosol vertical distribution and optical properties over China from long-term satellite and ground-based remote sensing"

_Atmospheric Chemistry and Physics, 2016_

## Referee Comment (RC1) · Anonymous Referee #3 · 25 Oct 2016

1. What are the differences between CALIOP and NIES lidar? 2. How about is the precision of CALIOP AOD in figure 3? There needs to compare CALIOP AOD with the ground AOD observation, e.g. AERONET AOD, in different regions. 3. In figure 6, there need increase the results in autumn and winter, not only in spring and in summer. And to analyze their differences. 4. In figure 7, how about is the result in autumn and winter? 5. Finally, the title is too large. The manuscript mainly investigated three sites' results and compared with the CALIOP. So the manuscript need greatly enrich the content to match the title.

---

## Referee Comment (RC2) · Anonymous Referee #2 · 31 Oct 2016

This paper describes aerosol climatology over China using CALIPSO/CALIOP. The method used in this paper is rather simple using CALIPO version 3 level 2 data, but the results are interesting and merit publishing in ACP. The paper is generally well-written. However, some of descriptions are not correct or not reasonable. Especially, previous works are not properly reviewed and some of the references are not original and not suitable.

Specific comments P 2 First paragraph: In my opinion, It is not appropriate to cite too many papers with a single simple statement. In the first sentence, citing Boucher et al.,

is reasonable, but it is not clear at all why He et al. and Peng et al. are cited here. The same thing for the following sentences.

P9 l.9 The sentence "The retrieved aerosol extinction coefficients suffer from large uncertainties ....." is miss leading. What about CALIOP level 2 data? Similar layer type classification and retrieval method using variable lidar ratio can be used for ground-based lidars. It is fine that the AOD-constrained retrieval method is used in this paper. But that is not clearly mentioned. That should be mentioned also in the caption of Fig. 2.

P9 l.12: It is not Huang et al. who first introduced the AOD-constrained Fernald method. The method was used already in 1994, for example, in Takamura et al, Appl. Opt. 33 (30) 7132-7140 (1994). If the AOD-constrained method was employed, it would be useful to present a histogram of the derived lidar ratio value.

P12, l.19: The volume depolarization ratio includes molecular scattering contribution. The discussion is consequently not very quantitative (though it is still useful). The definition in Eq. (4) is fine, and the contribution in the lower height is dominant. So the contribution of molecular scattering is probably not significant. The situation should be mentioned.

Figure 8: Definition of height should be provided. The profile with a large secondary peak in PRD MAM seems unusual as a climatological profile. What is the number of profiles averaged in this profile? The number of the data used and the error bars of the profiles should be presented. If the secondary peak is real, the source of aerosols in the secondary peak must be discussed. The descriptions in p. 18, l. 2-3 do not explain the cause of the secondary peak. Fan et al. paper is on the meteorological condition on October, not MAM. As to the vertical profile in Guangzhou, the following paper should be cited. It describes non-dust aerosol climatology in Beijing and Guangzhou using ground-based lidars and CALIOP. Hara et al., (2011) "Seasonal Characteristics of Spherical Aerosol Distribution in Eastern Asia: Integrated

Analysis Using Ground/Space-Based Lidars and a Chemical Transport Model" Scientific Online Letter on the Atmosphere, Vol. 7, 121−124, doi:10.2151/sola.2011-031 (https://www.jstage.jst.go.jp/article/sola/7/0/7_0_121/_article/)
* * *

---

## Referee Comment (RC3) · Anonymous Referee #1 · 11 Nov 2016

Using long-term satellite and ground-based remote sensing observations, this study describes the climatology of aerosol vertical distribution and optical properties over China, particularly for several important regions. In principle, this paper is well written and the findings are interesting.

The followings are my minor comments:

[Figure]

(1)Page 2, line 20, I would suggest to add references Garrett and Zhao 2006 and Zhao et al. 2015, which both show the strong warming climate effect of aerosols by serving as CCN and changing cloud properties. Garrett, T. J., C. Zhao, and P. C. Noel, 2010: Assessing the relative contributions of transport efficiency and scavenging to seasonal variability in Arctic aerosol. Tellus B, 62, 190-196. Zhao, C., and T. Garrett, 2015:Âă-Effects of Arctic haze on surface cloud radiative forcing,ÂăGeophys. Res. Lett.,Âă42, doi:10.1002/2014GL062015.

(2)Page 3, line 4, I would also suggest to add one reference which is about the effects of IN scheme representation using dust aerosols to radiation balance in climate model of CAM5. 21.Xie, S., X. Liu, C. Zhao, and Y. Zhang, 2013: Sensitivity of CAM5 simulated Arctic clouds and radiation to ice nucleation parameterization, J. Climate, 26, 5981–5999. doi:Âăhttp://dx.doi.org/10.1175/JCLI-D-12-00517.1.

(3)Page 3, line 16-18, is there any reference to support your claim that CALIOP AOD presents an underestimation because of the challenge of the thin layer detection. I am not sure if my understanding is right: if the thin layer clouds are missed, CALIOP AOD could be overestimated, not underestimated.

(4)Page 3, line 21, I would suggest "slight underestimation" instead of "small underestimation".

(5)Page 4, line 2, do you mean " seasonal averaged vertical profiles"?

(6)Page 4, line 4-6, what do you mean 'evaluate' here: do Guo et al. 2016a use satellite observations to evaluate the ground-based findings? The logic seems not right to me.

(7)Page 4, line 7, what do you mean for the "representative regions" here? Where are they?

(8)Page 5, line 3-4, I would suggest to add a reference Yang et al. 2016, which estimated the air pollution enhancement due to the aerosol-PBL feedback in Beijing. Yang, X., C. Zhao, J. Guo, and Y. Wang (2016), Intensification of aerosol pollution associated

with its feedback with surface solar radiation and winds in Beijing, J. Geophys. Res. Atmos., 121, 4093–4099, doi:10.1002/2015JD024645.

(9)Page 5, line 11, use 'between ... and ...' or ' the ratio of 1064 nm to 532 nm backscatter'

(10)Page 5, line 15-18, what are the major points or findings you want to use from this cited study?

(11)Page 6, line 1, ' have investigated' ->'investigate'.

(12)Page 6, line 8, 'in sections 4 and 5, respectively'.

(13)Page 6, line 16-17, are you sure that Northeast China Plain is one of the cleanest regions in China? I do not know if it is right but it seems that this region is often heavily polluted.

(14)Page 8, line 1, 'daytime solar background illumination' -> 'daytime background solar illumination'.

(15)Page 8, line 19, National Institute for Environmental Studies (NIES).

(16)Page 10, line 6, ' is set as a threshold value to define weakly ...'

(17)Page 11, line 3, how is the inconsistency, larger or smaller?

(18)Page 12, lines 7-8, this sentences have been repeated two times, you may just keep one time description.

(19)Page 16, line 5-9, for these findings or descriptions, may you please give the likely reasons?

(20)Page 17, line 14, ' is also correspond to ..' ->'corresponds to'

(21)Page 18, line 9, as suggested earlier, please add one reference by Yang et al. 2016.

(22)Page 18, line 12-13, is the claim generally right or just right for studied cases?

---

## Author Comment (AC1) · 23 Dec 2016

Using long-term satellite and ground-based remote sensing observations, this study describes the climatology of aerosol vertical distribution and optical properties over China, particularly for several important regions. In principle, this paper is well written and the findings are interesting. Response: The authors are grateful for the positive comments on our work. All the comments and concerns raised by the referee have been considered and incorporated into the revised manuscript. The followings are my minor comments: (1) Page 2, line 20, I would suggest to add references Garrett

and Zhao 2006 and Zhao et al. 2015, which both show the strong warming climate effect of aerosols by serving as CCN and changing cloud properties. Garrett, T. J., C. Zhao, and P. C. Noel, 2010: Assessing the relative contributions of transport efficiency and scavenging to seasonal variability in Arctic aerosol. Tellus B, 62, 190-196. Zhao, C., and T. Garrett, 2015:ÂËŸ a- Effects of Arctic haze on surface cloud radiative forcing,ÂËŸaGeophys. Res. Lett., ˟a42, doi:10.1002/2014GL062015. Response: The recommended references have been added in the revised manuscript. (2) Page 3, line 4, I would also suggest to add one reference which is about the effects of IN scheme representation using dust aerosols to radiation balance in climate model of CAM5. 21.Xie, S., X. Liu, C. Zhao, and Y. Zhang, 2013: Sensitivity of CAM5 simulated Arctic clouds and radiation to ice nucleation parameterization, J. Climate, 26, 5981–5999. doi:ÂËŸahttp://dx.doi.org/10.1175/JCLI-D-12-00517.1. Response: The recommended reference has been added in the revised manuscript. (3) Page 3, line 16-18, is there any reference to support your claim that CALIOP AOD presents an underestimation because of the challenge of the thin layer detection. I am not sure if my understanding is right: if the thin layer clouds are missed, CALIOP AOD could be overestimated, not underestimated. Response: The AOD underestimation of CALIOP has been discussed in several previous studies (e.g., Winker et al., 2013; Liu et al., 2014; Papagiannopoulos et a., 2016), which have been discussed in the revised manuscript. The ambiguous description "thin layer detection" has been replaced by "thin aerosol layer detection". The misclassification of thin layer clouds as aerosols leads to higher aerosol loading in the upper troposphere sometimes (Winker et al., 2013). However, the data quality of the CALIOP level 2 data is good enough in estimating the regional aerosol climatology (Yu et al., 2010; Winker et al., 2013; Amiridis et al., 2015). Those points have been provided in the revised manuscript. (4) Page 3, line 21, I would suggest "slight underestimation" instead of "small underestimation". Response: Modified as recommended. (5) Page 4, line 2, do you mean "seasonal averaged vertical profiles"? Response: We have added "average" in the sentence. (6) Page 4, line 4-6, what do you mean 'evaluate' here: do Guo et al. 2016a use satellite observations to evaluate the ground-based findings? The logic seems not right to me. Response: We have replaced the word "evaluated" with "estimated". Guo et al. (2016a) didn't use satellite observations to evaluate the ground-based findings. (7) Page 4, line 7, what do you mean for the "representative regions" here? Where are they? Response: The representative regions are the Taklimakan Desert, the Tibetan Plateau, the Loess Plateau, the Northeast China Plain, the Sichuan Basin, the North China Plain, the Yangtze River Delta, and the Pearl River Delta as defined in Section 2.1 and shown in Table 1 and Fig. 1. Each region represents one or more aerosol types such as dust, anthropogenic or mixed type aerosols. Those have been clarified in the revision.

(8) Page 5, line 3-4, I would suggest to add a reference Yang et al. 2016, which estimated the air pollution enhancement due to the aerosol-PBL feedback in Beijing. Yang, X., C. Zhao, J. Guo, and Y. Wang (2016), Intensification of aerosol pollution associated with its feedback with surface solar radiation and winds in Beijing, J. Geophys. Res. Atmos., 121, 4093–4099, doi:10.1002/2015JD024645. Response: The recommended reference has been added. (9) Page 5, line 11, use 'between ... and ...' or ' the ratio of 1064 nm to 532 nm backscatter' Response: Changed as recommended. (10) Page 5, line 15-18, what are the major points or findings you want to use from this cited study? Response: The cited study was removed because of less relevance. (11) Page 6, line 1, ' have investigated' ->'investigate'. Response: Modified as recommended. (12) Page 6, line 8, 'in sections 4 and 5, respectively'. Response: Modified as recommended. (13) Page 6, line 16-17, are you sure that Northeast China Plain is one of the cleanest regions in China? I do not know if it is right but it seems that this region is often heavily polluted. Response: The Northeast China Plain is one of the cleanest regions in the selected eight representative regions in China from the perspective of AOD in our research and in the previous studies (Luo et al., 2014; Tao et al., 2015). Some big cities may be heavily polluted sometimes in this region, but it presents less natural dust and anthropogenic pollution relative to the other selected regions except the Tibetan Plateau. Those points have been clarified in the revision. (14) Page 8, line 1, 'daytime solar background illumination' -> 'daytime background solar illumination'. Response: Modified as recommended. (15) Page 8, line 19, National Institute for Environmental Studies (NIES). Response: Modified as recommended. (16) Page 10, line 6, ' is set as a threshold value to define weakly ...' Response: Modified as recommended. (17) Page 11, line 3, how is the inconsistency, larger or smaller? Response: Smaller. The dusty conditions always show heavy aerosol load, so excluding profiles under such conditions decreases the values of the average extinction profiles from the CE370-2 lidar. We have modified the sentence in the revised manuscript. (18) Page 12, lines 7-8, this sentences have been repeated two times, you may just keep one time description. Response: We have removed the redundant sentence. (19) Page 16, line 5-9, for these findings or descriptions, may you please give the likely reasons? Response: We have modified our descriptions and provided the reasons in the revised manuscript. Strong winds transport boundary layer dust aerosols to higher altitudes in spring (Ge et al., 2016). Very low boundary layer height in the Taklimakan Desert region (Guo et al., 2016b) traps more aerosols near the surface. (20) Page 17, line 14, ' is also correspond to .' ->'corresponds to' Response: Modified as recommended. (21) Page 18, line 9, as suggested earlier, please add one reference by Yang et al. 2016. Response: The recommended reference has been added. (22) Page 18, line 12-13, is the claim generally right or just right for studied cases? Response: The claim is right in the representative regions over China, because this claim is based on climatological results from almost 10 years' CALIOP observations. We have attributed this phenomenon to the interactions between absorbing aerosols and the atmospheric boundary layer in the polluted regions. References Amiridis, V., Marinou, E., Tsekeri, A., Wandinger, U., Schwarz, A., Giannakaki, E., Mamouri, R., Kokkalis, P., Binietoglou, I., Solomos, S., Herekakis, T., Kazadzis, S., Gerasopoulos, E., Proestakis, E., Kottas, M., Balis, D., Papayannis, A., Kontoes, C., Kourtidis, K., Papagiannopoulos, N., Mona, L., Pappalardo, G., Le Rille, O., and Ansmann, A.: LIVAS: a 3-D multi-wavelength aerosol/cloud database based on CALIPSO and EARLINET, Atmos. Chem. Phys., 15, 7127-7153, doi:10.5194/acp-15-7127-2015, 2015. Liu, C., Shen X., Gao W., Liu P., and Sun Z.: Evaluation of CALIPSO aerosol

optical depth using AERONET and MODIS data over China, In SPIE Optical Engineering Applications 2014 Oct 2 (pp. 92210F-92210F), International Society for Optics and Photonics, doi:10.1117/12.2058929, 2014. Luo, Y.X., Zheng, X.B., Zhao, T.L., and Chen, J.: A climatology of aerosol optical depth over China from recent 10 years of MODIS remote sensing data, Int. J. Climatol. 34, 863-870, doi:10.1002/joc.3728, 2014. Papagiannopoulos, N., Mona, L., Alados-Arboledas, L., Amiridis, V., Baars, H., Binietoglou, I., Bortoli, D., D'Amico, G., Giunta, A., Guerrero-Rascado, J.L., Schwarz, A., Pereira, S., Spinelli, N., Wandinger, U., Wang, X., and Pappalardo, G.: CALIPSO climatological products: evaluation and suggestions from EARLINET, Atmos. Chem. Phys., 16, 2341-2357, 2016, doi:10.5194/acp-16-2341-2016, 2016. Tao, M., Chen, L., Wang, Z., Tao, J., Che, H., Wang, X., and Wang, Y.: Comparison and evaluation of the MODIS Collection 6 aerosol data in China, J. Geophys. Res., 120, 6992-7005, doi:10.1002/2015JD023360, 2015. Winker, D.M., Tackett, J.L., Getzewich, B.J., Liu, Z., Vaughan, M.A., and Rogers, R.R.: The global 3-D distribution of tropospheric aerosols as characterized by CALIOP, Atmos. Chem. Phys., 13, 3345-3361, doi:10.5194/acp-13-3345-2013, 2013. Yu, H., Chin, M., Winker, D.M., Omar, A.H., Liu, Z., Kittaka, C., and Diehl, T.: Global view of aerosol vertical distributions from CALIPSO lidar measurements and GOCART simulations: Regional and seasonal variations, J. Geophys. Res., 115, D00H30, doi:10.1029/2009JD013364, 2010.

---

## Author Comment (AC2) · 23 Dec 2016

This paper describes aerosol climatology over China using CALIPSO/CALIOP. The method used in this paper is rather simple using CALIPO version 3 level 2 data, but the results are interesting and merit publishing in ACP. The paper is generally well-written. However, some of descriptions are not correct or not reasonable. Especially, previous works are not properly reviewed and some of the references are not original and not suitable. Response: The authors are grateful for the helpful comments by this referee. All the comments and concerns raised by the referee have been considered

and incorporated into the revised manuscript.

Specific comments P2 First paragraph: In my opinion, It is not appropriate to cite too many papers with a single simple statement. In the first sentence, citing Boucher et al., is reasonable, but it is not clear at all why He et al. and Peng et al. are cited here. The same thing for the following sentences. Response: We have provided justifications for those relevant references, which have shown that the proper representation of mixing state is key to the assess the atmospheric stability because of black carbon particles.

P9 l.9 The sentence "The retrieved aerosol extinction coefficients suffer from large uncertainties ....." is miss leading. What about CALIOP level 2 data? Similar layer type classification and retrieval method using variable lidar ratio can be used for ground-based lidars. It is fine that the AOD-constrained retrieval method is used in this paper. But that is not clearly mentioned. That should be mentioned also in the caption of Fig. 2. Response: (1) This sentence has been removed from the revised manuscript. (2) CALIOP level 2 aerosol extinction coefficients suffer from uncertainties caused by the pre-assigned lidar ratios for certain aerosol types (Papagiannopoulos et al., 2016). However, the data quality of the CALIOP level 2 aerosol extinction coefficients is good enough in estimating regional aerosol climatology (Yu et al., 2010; Winker et al., 2013; Amiridis et al., 2015). (3) The AOD-constrained retrieval method has been mentioned in the caption of Fig. 2 in the revised manuscript. P9 l.12: It is not Huang et al. who first introduced the AOD-constrained Fernald method. The method was used already in 1994, for example, in Takamura et al, Appl. Opt. 33 (30) 7132-7140 (1994). If the AOD-constrained method was employed, it would be useful to present a histogram of the derived lidar ratio value. Response: (1) We have cited Takamura et al. (1994) for the AOD-constrained Fernald method. We have also revised our description to make it clearer to readers. (2) A histogram of the derived lidar ratio has been included in the supplement of the revised manuscript (Fig. S11).

P12, l.19: The volume depolarization ratio includes molecular scattering contribution. The discussion is consequently not very quantitative (though it is still useful). The

definition in Eq. (4) is fine, and the contribution in the lower height is dominant. So the contribution of molecular scattering is probably not significant. The situation should be mentioned. Response: This situation has been discussed in the revised manuscript.

Figure 8: Definition of height should be provided. The profile with a large secondary peak in PRD MAM seems unusual as a climatological profile. What is the number of profiles averaged in this profile? The number of the data used and the error bars of the profiles should be presented. If the secondary peak is real, the source of aerosols in the secondary peak must be discussed. The descriptions in p. 18, l. 2-3 do not explain the cause of the secondary peak. Fan et al. paper is on the meteorological condition on October, not MAM. As to the vertical profile in Guangzhou, the following paper should be cited. It describes non-dust aerosol climatology in Beijing and Guangzhou using ground-based lidars and CALIOP. Hara et al., (2011) "Seasonal Characteristics of Spherical Aerosol Distribution in Eastern Asia: Integrated Analysis Using Ground/Space-Based Lidars and a Chemical Transport Model" Scientific Online Letter on the Atmosphere, Vol. 7, 121-124, doi:10.2151/sola.2011-031 (https://www.jstage.jst.go.jp/article/sola/7/0/7_0_121/_article/) Response: (1) The definition of height has been added in the caption of Fig. 10 of the revised manuscript. (2) In our study, 3200 aerosol layers were detected by CALIOP in the PRD region in spring. The extinction coefficient of the detected aerosol layers were used to calculate an average profile. The average extinction profile with error bars in the PRD region in spring is shown in Fig. S12. (3) The profile with a large peak in the PRD MAM seems to be true, which has been proven by a recently published paper (Heese et al., 2016). Heese et al. (2016) used a multi-wavelengths Raman and depolarization lidar to observe aerosol vertical distribution at Sun Yat-sen University of Guangzhou in the PRD region. They found a lofted aerosol layer in the altitudes of 2 to 5 km in spring and characterized the aerosol type using the aerosol optical properties. They also used backward trajectory analysis to determine the origin and the sources of the lofted layers. They found that particles in the lofted aerosol layers in the PRD region are locally and regionally produced pollution mixtures. (4) The recommended reference has been cited in the revised manuscript. References Amiridis, V., Marinou, E., Tsekeri, A., Wandinger, U., Schwarz, A., Giannakaki, E., Mamouri, R., Kokkalis, P., Binietoglou, I., Solomos, S., Herekakis, T., Kazadzis, S., Gerasopoulos, E., Proestakis, E., Kottas, M., Balis, D., Papayannis, A., Kontoes, C., Kourtidis, K., Papagiannopou-los, N., Mona, L., Pappalardo, G., Le Rille, O., and Ansmann, A.: LIVAS: a 3-D multi-wavelength aerosol/cloud database based on CALIPSO and EARLINET, Atmos. Chem. Phys., 15, 7127-7153, doi:10.5194/acp-15-7127-2015, 2015. Heese, B., Baars, H., Bohlmann, S., Althausen, D., and Deng, R.: Continuous vertical aerosol profiling with a multi-wavelength Raman polarization lidar over the Pearl River Delta, China, Atmos. Chem. Phys. Discuss., 2016, 1-25, doi:10.5194/acp-2016-733, 2016. Papa-giannopoulos, N., Mona, L., Alados-Arboledas, L., Amiridis, V., Baars, H., Binietoglou, I., Bortoli, D., D'Amico, G., Giunta, A., Guerrero-Rascado, J.L., Schwarz, A., Pereira, S., Spinelli, N., Wandinger, U., Wang, X., and Pappalardo, G.: CALIPSO climatolog-ical products: evaluation and suggestions from EARLINET, Atmos. Chem. Phys., 16, 2341-2357, 2016, doi:10.5194/acp-16-2341-2016, 2016. Takamura, T., Sasano, Y., and Hayasaka, T.: Tropospheric aerosol optical properties derived from lidar, sun photometer, and optical particle counter measurements, Appl. Opt., 33, 7132-7140, doi:10.1364/AO.33.007132, 1994. Winker, D.M., Tackett, J.L., Getzewich, B.J., Liu, Z., Vaughan, M.A., and Rogers, R.R.: The global 3-D distribution of tropospheric aerosols as characterized by CALIOP, Atmos. Chem. Phys., 13, 3345-3361, doi:10.5194/acp-13-3345-2013, 2013. Yu, H., Chin, M., Winker, D.M., Omar, A.H., Liu, Z., Kittaka, C., and Diehl, T.: Global view of aerosol vertical distributions from CALIPSO lidar mea-surements and GOCART simulations: Regional and seasonal variations, J. Geophys. Res., 115, D00H30, doi:10.1029/2009JD013364, 2010.

---

## Author Comment (AC3) · 23 Dec 2016

The authors are grateful for the helpful comments from this referee. All the comments and concerns raised by the referee have been considered and incorporated into the revised manuscript. 1. What are the differences between CALIOP and NIES lidar? Response: The main difference between CALIOP and NIES lidar is the observation direction: the ground-based NIES lidar lies below the aerosol layers, with the emitted laser light penetrating the aerosol layers from the bottom to the top; while the satellite-based CALIOP lies above the aerosol layers, with the emitted laser light penetrating the

aerosol layers from the top to the bottom. The satellite-based CALIOP provides a global observation, while the NIES lidar provides continues observation over SACOL. The retrieval methods are different. There are also some differences in technical details as shown in Table S2. 2. How about is the precision of CALIOP AOD in figure 3? There needs to compare CALIOP AOD with the ground AOD observation, e.g. AERONET AOD, in different regions. Response: The precision of CALIOP AOD over China has been evaluated by Liu et al. (2014) using both AERONET and MODIS observations. They found that CALIOP AOD is lower than AERONET AOD. Better agreement is apparent at XiangHe, Beijing, Xinglong, and SACOL sites, while low correlations between CALIOP and AERONET observations were observed at Taihu and Hong_Kong_PolyU sites. Comparisons over China and other regions show that the overall spatial-temporal distribution of CALIOP AOD and MODIS AOD are basically consistent (Kittaka et al., 2011; Koffi et al., 2012; Liu et al., 2014). 3. In figure 6, there need increase the results in autumn and winter, not only in spring and in summer. And to analyze their differences. Response: The AERONET volume size distribution and spectral SSA in all seasons have been included in Fig. 7 in the revised manuscript, and the seasonal variation has been also discussed. In Fig. 6 (Fig. 8 in the revised manuscript), we have selected a few sites to better compare the dust-dominant aerosols, anthropogenic aerosols and mixed type aerosols. Beijing, SACO,L and Taihu sites were selected because of large data amount at these sites (Table S1). SACOL is dominated by dust aerosols in spring, Beijing and Taihu are dominated by anthropogenic aerosols in summer, and Beijing represents mixed type aerosols by dust and anthropogenic pollution in spring. Therefore, we only selected SACOL in spring, Beijing and Taihu in summer, and Beijing in spring. 4. In figure 7, how about is the result in autumn and winter? Response: Seasonal aerosol properties from the ground-based AERONET observations have been studied using the volume size distribution and spectral SSA (Fig. 7 in the revised manuscript). In Fig. 7 (Fig. 9 in the revised manuscript), we selected a few sites to better compare the dust-dominant aerosols, anthropogenic aerosols and mixed type aerosols. As explained in Comment 3, we only selected SACOL in spring,

Beijing and Taihu in summer, and Beijing in spring. 5. Finally, the title is too large. The manuscript mainly investigated three sites' results and compared with the CALIOP. So the manuscript need greatly enrich the content to match the title. Response: We have enriched the content by including as additional AERONET observations over China. All the AERONET sites with an observation of more than 3 months in the representative regions were selected (Fig. S1 and Table S1). As a result, 17 sites in the Loess Plateau, the North China Plain, the Pearl River Delta, the Tibetan Plateau, and the Yangtze River Delta regions were included in our study. In addition, 4 desert sites in Hexi Corridor of Gansu in northwest China were selected to represent dust aerosols. The climatological results were included in the revised manuscript (Figs. 6 and 7).

References Kittaka, C., Winker, D. M., Vaughan, M. A., Omar, A., and Remer, L. A.: Intercomparison of column aerosol optical depths from CALIPSO and MODIS-Aqua, Atmos. Meas. Tech., 4, 131-141, 10.5194/amt-4-131-2011, 2011. Koffi, B., Schulz, M., Bréon, F.-M., Griesfeller, J., Winker, D., Balkanski, Y., Bauer, S., Berntsen, T., Chin, M., Collins, W.D., Dentener, F., Diehl, T., Easter, R., Ghan, S., Ginoux, P., Gong, S., Horowitz, L.W., Iversen, T., Kirkevåg, A., Koch, D., Krol, M., Myhre, G., Stier, P., and Takemura, T.: Application of the CALIOP layer product to evaluate the vertical distribution of aerosols estimated by global models: AeroCom phase I results, J. Geophys. Res., 117, D10201, doi:10.1029/2011JD016858, 2012. Liu, C., Shen X., Gao W., Liu P., and Sun Z.: Evaluation of CALIPSO aerosol optical depth using AERONET and MODIS data over China, In SPIE Optical Engineering Applications 2014 Oct 2 (pp. 92210F-92210F), International Society for Optics and Photonics, doi:10.1117/12.2058929, 2014.